# Microbiological profile of diabetic foot ulcers in Kuwait

**Asma Alhubail[1], May Sewify[1], Grace Messenger[1], Richard Masoetsa[1], Imtiaz Hussain[1], Shinu Nair[1], Ali Tiss [2]***

1 Medical Division, Dasman Diabetes Institute, Kuwait City, Kuwait, 2 Biochemistry & Molecular Biology Department, Research Division, Dasman Diabetes Institute, Kuwait City, Kuwait

* ali.tiss@dasmaninstitute.org

## Abstract

### Introduction

Diabetic foot ulcers (DFU) and infection (DFI) are a major diabetes-related problem around the world due to the high prevalence of diabetes in the population. The aim of our study was to determine the microbiological profile of infected ulcers in patients attending Dasman Diabetes Institute (DDI) clinics in Kuwait and to analyze the distribution of microbial isolates according to wound grade, sex, age and diabetes control.

### Methods

We collected and analyzed clinical data and samples from 513 diabetic patients with foot ulcers referred to our podiatry clinic at DDI from Jan 2011 till Dec 2017.

### Results

We show a higher prevalence of DFU in men than in women, and a greater percentage of DFU occurred in men at an earlier age (p<0.05). Only about half of the DFU were clinically infected (49.3%) but 92% of DFU showed bacterial growth in the microbiological lab analysis. In addition, we isolated more monomicrobial (57.3%) than polymicrobial (34.8%) DFI and representing an average of 1.30 pathogens per patient. The presence of Gram-positive and Gram-negative strains was comparable between men and women regardless their age or glucose levels. Interestingly, more Gram-positive strains are present in ulcers without ischemia while more Gram-negative strains are present in ulcers with ischemia (p<0.05). While *Staphylococcus aureus* was common in infected ulcers without ischemia, *Pseudomonas aeruginosa* was predominant in ulcers with infection and ischemia, regardless of ulcer depth. Finally, a higher percentage of women has controlled HbA1c levels (19.41% versus 11.95% in men) and more women in this group displayed non-infected wounds (60.6% and 43.90% for women and men, respectively).

### Conclusion

Our results provide an updated picture of the DFI patterns and antibiotics resistance in patients attending Dasman Diabetes Institute (DDI) clinics in Kuwait which might help in adopting the appropriate treatment of infected foot and improving clinical outcomes.

**Data Availability Statement:** All relevant data are within the paper and its Supporting information files.

**Funding:** No specific funding was requested or provided for this study as all analyses were

performed by our staff within our institute using available electronic data.

**Competing interests:** The authors have declared that no competing interests exist.

**Abbreviations:** CAP, College of American Pathologists; DDI, Dasman Diabetes Institute; DFI, Diabetic Foot Infection; DFU, Diabetic Foot Ulcer; MRSA, Methicillin-resistant *Staphylococcus aureus*.

# 1 Introduction

Diabetic foot ulcers (DFU) are one of the commonest long-term complications of diabetes mellitus. The lifetime risk of developing a DFU in diabetic patients is estimated to be 12 to 25% [1–3]. According to the "Diabetic Foot Ulcers-Epidemiology Forecast to 2025," which includes data from seven countries, over 1 million foot-ulceration cases were reported in 2015, and the estimated increasing rate is higher than 4% per year in those countries [4]. Despite the fact that the prevalence of diabetes in Kuwait (22.0%) is ranked among the top 20 highest worldwide [5], a single centered study conducted in Kuwait reported an incidence of 3.4% for DFU [6]. The etiology of DFU is multi-factorial with the triad of diabetic peripheral neuropathy, peripheral arterial disease, and foot deformity responsible for the majority of DFU. Another major contributing factor to DFU outcomes is diabetic foot infection (DFI), which has been found to be present in 40–60% of all DFU [7, 8], and is associated with increased hospital admissions, worsening outcomes, and increased amputation rates [9, 10].

Clinically, DFI is said to be present if there are two or more cardinal signs of inflammation (induration, erythema, raised temperature, increased pain, and purulent discharge) [9]. DFI can be categorized as mild, moderate, or severe and are frequently polymicrobial, with multiple bacteria identified in them [9, 11]. Gram-positive bacteria, such as *Staphylococcus aureus*, and Gram-negative bacteria, such as *Pseudomonas aeruginosa*, are the most common pathogens [9]. However, the incidence and the prevalence of Methicillin Resistant *Staphylococcus aureus* (MRSA) have both increased, which has been associated with improper antibiotic use and non-restrictive regulations controlling antibiotic abuse [12].

The diversity of bacteria in chronic wounds is considered an important contributor to the chronicity and severity of DFU [11], and the polymicrobial nature of DFI has been well studied [9, 13–15]. However, results were variable. Studies have found that *Staphylococcus aureus* is the main causative pathogen [16–18], but others have reported a predominance of Gram-negative aerobes [19, 20]. These discrepancies could be related to various factors, such as the presence of different causative organisms at different times, geographical variations and ecologies, socio-economic conditions, hygiene, accessibility to effective health care services, the type and the severity of the infections, or depth of the wound and sampling technique [14, 21]. These differences in microbial pathogens have triggered considerable efforts to decipher their impact on DFI and improve treatment outcomes [22]. Consequently, studies have shown that parenteral and oral antibiotic treatment, guided by culture, successfully prevent amputations in a large proportion of patients with diabetes admitted with a DFU or suspected osteomyelitis [23].

In order to obtain the evidence necessary to enhance the effectiveness of DFI treatments in Kuwait, we aimed to: (i) profile and compare the microbes isolated from DFI from different ulcer grades using the University of Texas Wound Classification and combine these findings with antimicrobial sensitivity (ii) identify the most frequent DFI pathogens within our geographical region (iii) determine if the microbial profile of DFI differ according to patients age, diabetes control and sex. We retrospectively analyzed clinical data and microbiological samples collected from 513 patients with DFU treated at the Podiatry Department at the Dasman Diabetes Institute (DDI) in Kuwait for a period of 7 years.

# 2 Materials and methods

## 2.1 Study population

We collected data for this study from electronic health records at the Podiatry department, Dasman Diabetes Institute (DDI) from January 2011 to December 2017. DDI is a specialized

outpatient center helping patients with diabetes mellitus manage their blood glucose levels and treat diabetes complications. Diabetic patients with complications were referred to DDI from all governorates in Kuwait and might be considered representative of all Kuwait population. During this period, 835 patients with DFU were referred to the Podiatry department, this represented around 120 patients per year. Microbiology samples were taken from 513 patients within this cohort at their first visit to our podiatry services. We obtained information on patients' age, sex, diabetes control, and treatment history of diabetes from the DDI Laboratory Information System (LIS) and the main parameters are summarized in Table 1. Approval and a waiver from the need to provide written informed consent were obtained from The Ethical Review Board of DDI for the access and use of anonymized data from the LIS for the purpose of publication.

## 2.2 Wound grading

Our experienced podiatrists classified the DFU according to the University of Texas Wound (UoT) Grading Classification System which grades DFU by depth (0, I, II, III) and stage (A, B, C, D) depending on the presence or absence of infection and ischemia [24]. This comprehensive and validated classification tool was consistently used for all subjects to develop a management plan and determine the eventual outcome of the DFU which are known to deteriorate with increased stages and grades. Clinical diagnosis of infection which has previously been defined [9, 23] by the presence of at least 2 of the following indicators: local swelling or induration, >0.5 cm of erythema around the wound, local tenderness or pain, local warmth, and purulent discharge was used.

## 2.3 Samples collection and processing

Samples were taken by the podiatry staff at the patient's first appointment as per DDI policy at the time, or when there was clinical suspicion of DFI. Prior to sample collection the DFU was cleansed with sterile saline solution, and sharp debrided. Samples were collected either using a surface swab or wound biopsy. Surface swabs were collected using Levine's Technique [25], involving the rotation of a wound swab over a 1 $cm^2$ area of the wound for 5 seconds, using sufficient pressure to extract fluid from the inner part of the wound. Deep tissue specimens of

**Table 1. Study population characteristics.**

| Diabetes Status | Parameter | Sex | | Total |
|---|---|---|---|---|
| | | **Male** | **Female** | |
| Controlled (HbA1c<6.5%) | N = | 41 | 33 | 74 |
| | | (11.95%) [a] | (19.41%) [b] | (14.42%) [c] |
| | age (years) | 60 ± 12 | 64 ± 9 | 62 ± 11 |
| Uncontrolled (HbA1c ≥6.5%) | N = | 302 | 137 | 439 |
| | | (88.05%) [a] | (80.59%) [b] | (85.58%) [c] |
| | Age (years) | 61 ± 11 | 65 ± 10 | 62 ± 11 |
| **Total** | N = | **343** | **170** | **513** |
| | | (66.86%) [c] | (33.14%) [c] | |
| | Age (years) | **61 ± 11** | **65 ± 10** [*] | **62 ± 11** |

[*]: *p<0.05 between sexes.*

[a] (% from all Males)

[b] (% from all Females)

[c] (% from all population)

4mm in diameter were obtained from the base of the ulcer during the sharp debridement process. The specimens were placed into sterile transport containers and sent to the microbiology laboratory for aerobic culturing within 20 minutes. Anaerobic culturing was not performed in this study.

Different types of media were used for sample inoculations; namely, blood, MacConkey, chocolate, mannitol-salt (MSA), Sabouraud agars, and thioglycolate broth (Oxoid, Basingstoke, UK). All plates were incubated for a maximum of five days or until visible growth using aerobic ($O_2$) at 37˚C, which ever come first; except for MRSA plates, which were incubated at 32˚C. Cultures with any volume of bacterial growth present on at least one of the inoculated plates was reported as laboratory positive. We did not consider lower or upper bacterial growth limits when reporting the presence of bacterial growth.

## 2.4 Identification of isolated microorganisms

All microbiological analyses were performed in our College of American Pathologist- (CAP) accredited Clinical lab at DDI (CAP Lab #8086466). Cultures were processed following the same CAP approved standard procedures for both swab and tissue samples and identification was done according to the Clinical & Laboratory Standards Institute (CLSI -M50 and CLSI-M100). All plates were examined for visible growth after 24 hours of incubation. Subcultures for thioglycolated broth tubes were done after a 24-hour incubation period, using blood, MacConkey, and Sabouraud agar plates. Wound smears from each sample were Gram-stained to identify if Gram-positive or Gram-negative isolates were present. We considered no-growth plates as sterile after five days of incubation. For positive swab or tissue cultures, we performed identifications using an automated Microscan system (Walkaway 40 SI, Siemens Healthcare Diagnostics, and Sacramento, CA). We obtained panels for Gram-negative (NC34, NC50 and NC53) and for Gram-positive bacteria (PC21, PC39) from Siemens Healthcare Diagnostics (Sacramento, CA). For confirmation, we performed further biochemical tests for both Gram-positive and Gram-negative isolates using commercially available reagents (API 20E, API strep, and API staph) supplied by BioMérieux (Durham, NC, USA). MRSA confirmation was done at the Faculty of Medicine services, Kuwait University, by DNA sequencing (MicoSeq) of the isolates. Quality control strains (*Escherichia coli* ATCC® 25922, *Klebsiella pneumonia* ATCC® 700603, *Candida albicans* ATCC® 10231, *Pseudomonas aeruginosa* ATCC 27853, *Staphylococcus aureus* ATCC 29213, *Enterococcus faecalis* ATCC 29212) were supplied by the American Type Culture Collection (ATCC) (Manassas, VA).

## 2.5 Antimicrobial susceptibility testing

Susceptibilities to selected antibiotics for the common isolated bacteria (*Staphylococcus aureus*, *Pseudomonas aeruginosa*, *Klebsiella pneumoniae*, *Escherichia coli*, *Acinetobacter* sp *and Enterobacter* sp) were examined. Antimicrobial sensitivity testing of all isolates was performed on diagnostic sensitivity test plates according to the Kirby-Bauer method [26] following the definition of the Clinical and Laboratory Standards Institute (CLSI, 2014). Antibiotics and their concentrations used for antimicrobial susceptibility testing, are as following: Amikacin 30ug, Ampicillin10ug, Co-Amoxi/clav30 ug, Ceftazidime 30ug, Ciprofloxacin 5ug, Clindamycin 2ug, Ceftriaxone 30ug, Cefotaxime 30ug, Erythromycin 15ug, Gentamicin 10ug, Imipenem 10ug, Linezolid 30ug, Meropenem 10ug, Piperacillin/Tazobactam 110ug, Trimethoprim-Sulfamethoxazole 25ug, and Vancomycin 30ug. Bacterial inoculums were prepared by suspending the freshly grown bacteria in 5mL sterile saline solution. A sterile cotton swab was used to streak the surface of Mueller Hinton agar plates (Oxoid, Basingstoke, UK). Filter paper disks

containing a designated concentration of the antimicrobial drugs obtained from Becton and Dickinson Company (Franklin Lakes, NJ) were used.

## 2.6 Statistical analysis

We analyzed the data using the statistical SPSS software, v22.0 (IBM SPSS Statistics for Windows, IBM Corp. Armonk, NY, USA). We evaluated differences between continuous and categorical variables using the Student's t-test and the chi-square with F-test, respectively. We calculated correlations between variables with the Spearman's rank correlation test. All statistical assessments were two sided and considered significant at p values <0.05.

## 3 Results

### 3.1 Study population characteristics

During the study period, 513 patients attended the Podiatry Department at DDI with a diabetic foot ulcer which underwent laboratory investigation following swab or tissue sample collection. The mean age of patients was 62 ± 11 years (95% CI, 61.7–63.5), with a mean HbA1c of 8.6 ± 1.9% (95% CI, 8.4–8.8, equivalent to 70mmol/mol). The study group contained twice as many men as women (66.9% vs. 33.1%, respectively), with a statistically significant difference in the mean age between sexes (61 ± 11; 95% CI, 60.5–62.8 in men and 65 ± 10; 95% CI, 63.1–66.3 in women; p < 0.001,). However, we found no significant difference in mean HbA1c values between sexes (8.7± 1.9%; 95% CI, 8.5–8.9 for men and 8.4 ± 1.9%; 95% CI 8.1–8.7 for women; p = 0.178). In our study population, 85.6% of patients had uncontrolled HbA1c levels (≥6.5% or 48 mmol/mol), among them 22.8% had HbA1c ≥10% (equivalent to 86 mmol/mol) (Table 1). When dividing data according to HbA1c levels, we found more women (19.41%) than men (11.95%) with controlled HbA1c levels (p = 0.024).

Using the University of Texas (UoT) Wound Classification System, during the podiatry appointment, at the time of sample collection, we classified 253 (49.3%) DFU as clinically infected. This included 30.4% as UoT grade B (infection only) and 18.9% as UoT grade D (infection and ischemia) from the total study population. The remaining 260 patients (50.7%) had no clinical signs or symptoms of infection, including 42.5% as grade A and only 8.2% as grade C (Table 2). Within the whole study population just over one-quarter (27.1%) of ulcers were complicated with ischaemia, of which 69.7% also had infection. There was no difference in sex between the two groups, with neither sex being more likely to present with an infected ulcers than the other (49.6% and 48.8% of males and female, respectively; p = 0.642) (Table 2 & Fig 1A). Similarly, there was no difference in means between age (p = 0.650) or HbA1c (p = 0.310). In addition, when splitting the two groups depending on their controlled and uncontrolled HbA1c levels, no significant difference was observed

**Table 2. Texas wound grading distribution among the study population.**

| | Sex | HBA1c Controlled (<6.5%) | HBA1c Uncontrolled (≥6.5%) | Total |
|---|---|---|---|---|
| **Texas wound grads A&C** | male | 18 | 155 | 173 |
| (no *infection ± ischemia*) | female | 20 | 67 | 87 |
| *(Control Group)* | *Total* | **38** | **222** | **260** |
| **Texas wound grads B&D** | male | 23 | 147 | 170 |
| (*infection ± ischemia*) | female | 13 | 70 | 83 |
| | *Total* | **36** | **217** | **253** |
| **Total** | | **74** | **439** | **513** |

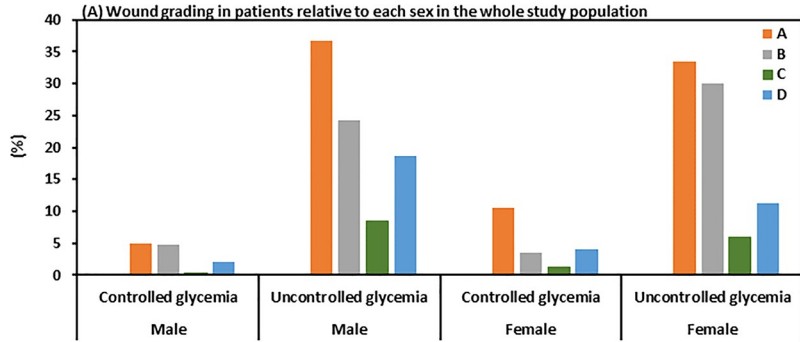

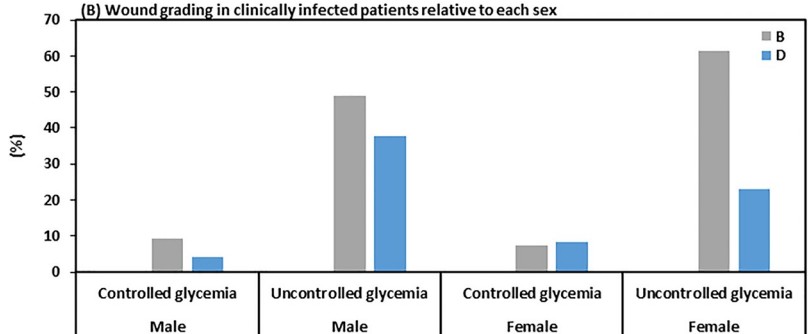

**Fig 1. Wound grading distribution among the DFU study population.** Patient DFU were classified according to the University of Texas Wound Grading Classification System which grades them depending on the presence or absence of infection and ischemia, as detailed in Material and Methods section.

between the groups (Fig 1A & 1B). Nevertheless, there was significantly more males with deeper and more complex DFU than females ($\chi 2(2) = 6.135$; $p = 0.047$). Interestingly, there was a significant difference between the mean age and ulcer classification; patients with infected ulcers were approximately 5 years younger than patients with infection and ischemia ($60.3\pm10.7$ and $65.9\pm13.4$ respectively, $p <0.001$). Additionally, this trend was continued when we took into consideration depth of DFU ($p = 0.002$).

There were 207 clinically uninfected (Grade A & C) and 196 clinically infected (Grade B & D) DFU with HbA1c levels below 10.0% and 53 uninfected and 57 infected FDUs with HbA1c greater than 10.0%. However, there was no significant difference in the distribution of HbA1c levels between infected and uninfected DFU classification ($p = 0.455$) (Table 2). Using Chi-square analysis, no significant association was observed between diabetes control and the presence of infection ($\chi 2(2) = 0.3503$; $p = 0.554$). Nevertheless, when we grouped the infected and uninfected DFU by UoT depth, a significant difference in mean HbA1c was found ($F(5, 246) = 2.584$, $p = 0.027$). Nearly 51% of DFU in our patients had no clinical signs of infection and were graded as UoT A or C. Interestingly, 47.7% of grade A and 42.9% of grade C DFU grew at least one of the following bacteria, Methicillin Sensitive *Staphylococcus aureus* (MSSA), MRSA, and *Pseudomonas aeruginosa* (data not shown).

## 3.2 Microbial isolate etiologies in ulcers with clinical infection

To better characterize the microbial etiology related to DFI, in the remaining part of our study and analyses we only included the clinically infected wounds (n = 253). Tables 3 and 4 summarize the distribution of microorganism isolates among the study population with

**Table 3. Distribution of microorganisms isolated from various samples used in the study.**

| Isolates Name | Wound swab Culture | Tissue Culture | Bone Culture | Total (% of frequency) |
|---|---|---|---|---|
| *Staphylococcus aureus* | 116 | 7 | 2 | **125 (17.9%)** |
| Coagulase negative *Staphylococcus* sp | 117 | 12 | 1 | **130 (18.5%)** |
| *Pseudomonas aeruginosa* | 80 | 15 | 7 | **102 (14.6%)** |
| *Klebsiella* sp | 45 | 8 | 4 | **57 (8.1%)** |
| *Citrobacter* sp | 14 | 0 | 1 | **15 (2.14%)** |
| *Enterobacter* sp | 45 | 6 | 1 | **52 (7.4%)** |
| *Streptococcus* sp | 27 | 4 | 0 | **31 (4.4%)** |
| *Candida albicans* | 4 | 0 | 0 | **4 (0.6%)** |
| *Morgenella* sp | 20 | 0 | 0 | **20 (2.9%)** |
| *Protues* sp | 29 | 1 | 1 | **31 (4.4%)** |
| *Acinitobacter* sp | 14 | 0 | 0 | **14 (2.0%)** |
| *Serratia* sp | 5 | 0 | 0 | **5 (0.71%)** |
| *Achromobacter xylosoxidans* | 1 | 0 | 0 | **1 (0.14%)** |
| *Ochrobacterium anthropi* | 1 | 0 | 0 | **1 (0.14%)** |
| *Escherichia coli* | 36 | 5 | 2 | **43 (6.1%)** |
| *Enterococcus faecalis* | 20 | 1 | 0 | **21 (3%)** |
| MRSA | 44 | 3 | 0 | **47 (6.7%)** |
| **Total** | **618** | **62** | **19** | **699** |

clinical infection. Ninety-two percent of all samples collected grew one or more isolates. In these patients with infected ulcers, the mean age was significantly higher in those without growth (66.4 ± 9.2; CI, 63.6–69.3 years) than those with bacterial growth (61.3 ± 10.9; CI, 60.3–63.3 years); t(53.2) = −2.96, p = 0.005. However, we found no such significance in sex or HbA1c between those two groups (p = 0.108). Most microbiological samples were

**Table 4. Microorganism isolates distribution among the study population with clinically infected wounds.**

| Isolate name | B1 | B2 | B3 | Total B | D1 | D2 | D3 | Total D | Total |
|---|---|---|---|---|---|---|---|---|---|
| *Staphylococcus aureus* | 29 | 3 | 5 | **37 (66.1%)** | 8 | 1 | 10 | **19 (33.9%)** | 56 (19.9%) |
| *Coagulase negative staphylococcus* | 24 | 1 | 8 | **33 (75%)** | 7 | 0 | 4 | **11 (25%)** | 44 (15.7%) |
| *Streptococcus* sp | 6 | 2 | 5 | **13 (81.3%)** | 0 | 1 | 2 | **3 (18.7%)** | 16 (5.7%) |
| *Enterococcus fecalis* | 1 | 1 | 4 | **6 (50.0%)** | 0 | 0 | 6 | **6 (50.0%)** | 12 (4.3%) |
| MRSA | 3 | 2 | 7 | **12 (85.7%)** | 0 | 0 | 2 | **2 (14.3%)** | 14 (5.0%) |
| *Pseudomonas aeruginosa* | 5 | 2 | 7 | **14 (38.9%)** | 7 | 3 | 12 | **22 (61.1%)** | 36 (12.8%) |
| *Klebsiella* sp | 6 | 3 | 0 | **9 (39.1%)** | 7 | 2 | 5 | **14 (60.9%)** | 23 (8.2%) |
| *Citrobacter* sp | 5 | 0 | 1 | **6 (75.0%)** | 0 | 1 | 1 | **2 (25%)** | 8 (2.8%) |
| *Acinetobacter* sp | 3 | 0 | 2 | **5 (100%)** | 0 | 0 | 0 | **0** | 5 (1.8%) |
| *Enterobacter* sp | 5 | 1 | 6 | **12 (54.5%)** | 2 | 1 | 7 | **10 (45.5%)** | 22 (7.8%) |
| *Morgenella* sp | 2 | 1 | 1 | **4 (57.1%)** | 2 | 0 | 1 | **3 (42.9%)** | 7 (2.5%) |
| *Proteus* spp | 4 | 0 | 5 | **9 (60.0%)** | 1 | 1 | 1 | **3 (40.0%)** | 12 (4.3%) |
| *Serratia* spp | 0 | 0 | 0 | **0** | 0 | 0 | 1 | **1 (100%)** | 1 (0.3%) |
| *Achromobacter xylosoxidans* | 1 | 0 | 0 | **1 (100%)** | 0 | 0 | 0 | **0** | 1 (0.3%) |
| *Ochrobactrum anthropi* | 0 | 0 | 1 | **1 (100%)** | 0 | 0 | 0 | **0** | 1 (0.3%) |
| *Escherichia coli* | 5 | 0 | 4 | **9 (45.0%)** | 4 | 2 | 5 | **11 (55.0%)** | 20 (7.1%) |
| *Candida albicans* | 0 | 0 | 0 | **0** | 0 | 1 | 2 | **3 (100%)** | 3 (1.0%) |
| **Total** | **99** | **16** | **56** | **171** | **38** | **13** | **59** | **110** | **281** |

monomicrobial (57.3%), followed by polymicrobial (34.8%) and no growth (7.9%). No significant association between sex or diabetes control and number of isolates (p = 0.08 and p = 0.403, respectively) was found.

The presence of Gram-positive and Gram-negative strains was comparable (50.5% -including 5.0% MRSA- and 48.4%, respectively), while yeast was present only in 1.1% of cultures (Table 4). There was no significant association between sex and Gram type (p = 0.801). Significant trends were, however, observed between DFU depth and bacteria groups. For instance, superficial infected ulcers not involving tendon, capsule or bone and DFU penetrating to tendon or capsule were more likely to be infected by gram positive bacteria than those of the same depth and complicated by the presence of infection and ischemia ($\chi 2$ = 5.65 (df2), p = 0.0175 and $\chi 2$ (2)23.376, p <0.001, respectively). As shown in Table 4, most Gram-positive isolates were *Staphylococcus aureus* (19.9%), coagulase-negative *Staphylococcus aureus* (15.7%), *Streptococcus* spp. (5.7%), MRSA (5.0%) and *Enterococcus faecalis* (4.3%). Within the Gram-negative isolates, the spectrum of strains was larger, and the main isolates were *Pseudomonas aeruginosa* (12.8%), *Klebsiella* species (8.2%), *Enterobacter* species (7.8%), *Escherichia coli* (7.1%) and *Proteus* spp. (4.3%).

We further analyzed the number of isolates present in those ulcers with clinical infection but different depths (I, II, III), however, no association was found between these parameters ($\chi 2(7)$ = 6.27; p = 0.180). In contrast, *Staphylococcus aureus* and MRSA were predominantly found in infected ulcers while *Pseudomonas aeruginosa* was predominantly found in ulcers with infection and ischemia as reflected by the number of patients in each of the DFU grades (Table 4) and the relative distribution of these 3 strains within each of those grades (Fig 2) (p<0.05). Indeed, *Staphylococcus aureus* and MRSA were isolated from 66.1% and 85.7% of

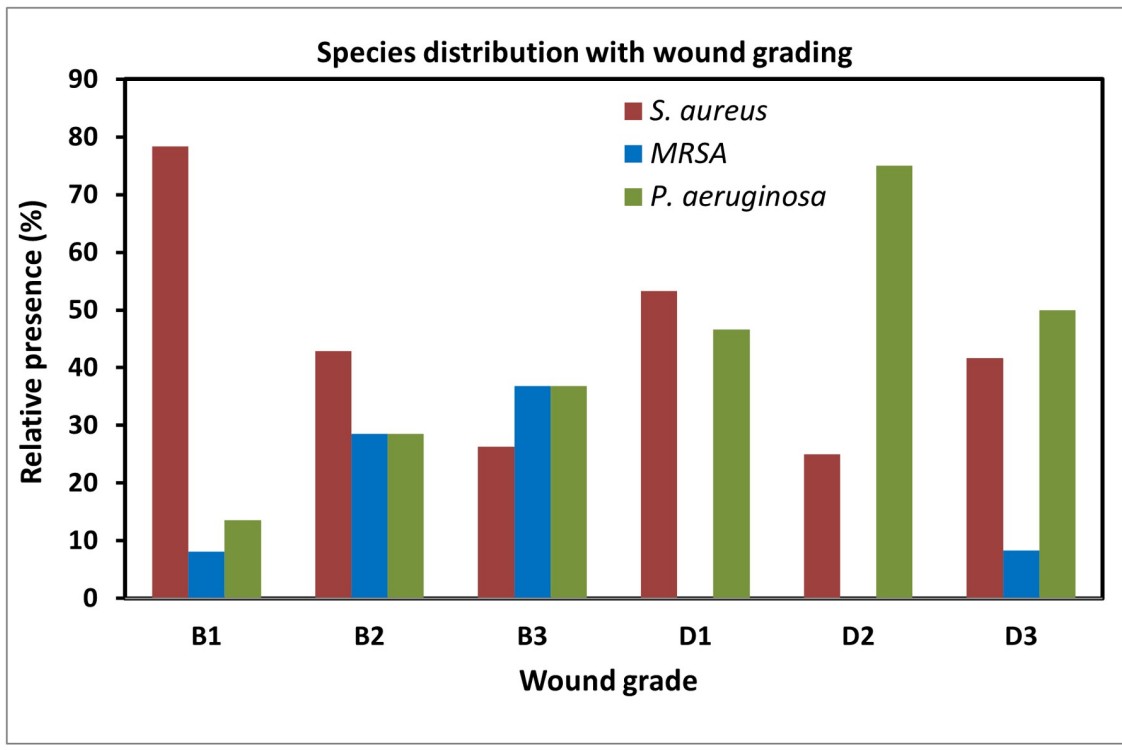

**Fig 2. Main species distribution among the study population with clinically infected wounds.** The species distribution was calculated as percentage of the total number of isolates within the same wound grade.

**Table 5. Antibiogram sensitivity pattern for most prevalent isolates.**

*Table 5a: Gram positive isolates*

| Organism Isolated | AMC | AMP | ERY | DA | VA | CIP | GM | LZ | | | |
|---|---|---|---|---|---|---|---|---|---|---|---|
| *Staphylococcus aureus* | 64% | 17% | 46.4% | 56.6% | 100% | 73% | 78% | 97.5% | | | |

*Table 5b: Gram positive isolates*

| Organism Isolated | AMC | AMP | AK | SXT | CRO | CTX | CIP | CAZ | TZP | GM | MEM | IMP |
|---|---|---|---|---|---|---|---|---|---|---|---|---|
| *Pseudomonas aeruginosa* | - | - | 87% | - | - | - | 77.2% | 65.7% | 71.1% | 80% | 88.4% | 85% |
| *Klebsiella pneumonia* | 60% | - | 88.6% | 61% | 65.8% | 63.6% | 77.2% | 63.6% | 93% | 91% | 100% | 100% |
| *Escherichia coli* | 56% | 19.4% | 100% | 39.4% | 48.6% | 68.4% | 63.2% | 73.6% | 100% | 97.3% | 100% | 100% |
| *Acinetobacter sp* | - | - | 100% | 91.6% | 33.3% | 33.3% | 61.5% | 75% | 83.3% | 92.3% | 100% | 100% |
| *Enterobacter sp* | - | - | 94.5% | 89% | 31.8% | 60% | 85.2% | 63.1% | 76.3% | 94.5 | 96.3% | 96.3% |

AK: Amikacin; AMP: Ampicillin; AMC: Co-Amoxiclav; CAZ: Ceftazidime; CIP: Ciprofloxacin; DA: Clindamycin; CRO: Ceftriaxone; CTX: Cefotaxime; ERY: Erythromycin; GM: Gentamicin; IMP: Imipenem; LZ: Linezolid; MEM: Meropenem; TZP: Piperacillin/ Tazobactam; SXT; Trimethoprim-Sulfamethoxazole; VA: Vancomycin

infected ulcers as compared to 33.9% and 14.3% respectively in DFU with infection and ischemia, respectively, whereas *Pseudomonas aeruginosa* was present 61.1% of DFU with infection and ischemia compared to only 38.9% in infected ulcers (Table 4). In the next step, we analyzed the potential association between bacterial strain and depth (I, II, III). DFU penetrating to bone or joint (BIII or DIII) had a higher proportion of MRSA in comparison to DFU with less depth, however, the limited number of samples in some sub-groups did not allow us to draw a clear conclusion on the association of bacterial strains within DFU subgrades (Fig 2).

## 3.3 Antimicrobial susceptibility pattern

The most prevalent gram-negative and gram-positive pathogens isolated from our DFI patients were tested for their sensitivity against a wide range of antibiotics and results are shown in Table 5. *Staphylococcus aureus* showed low resistance to vancomycin and linezolid (0% and 2.5%, respectively) but was more resistant to other tested antibiotics (53.6% and 83% for erythromycin and ampicillin, respectively). Of the gram-negative isolates, *Pseudomonas aeruginosa* was relatively less resistant to meropenem, amikacin, imipenem, gentamicin, and ciprofloxacin (11.6%, 13%, 15%, 20% and 22.8%, respectively). Meanwhile, *Escherichia coli* strains were not resistant or showing very low resistance to imipenem, meropenem, amikacin, piperacillin/ tazobactam and gentamicin (0%, 0%, 0%, 0% and 2.7%, respectively). *Klebsiella* species were not or barely resistant to imipenem, meropenem, piperacillin/tazobactam, and gentamicin (0%, 0%, 7% and 9%, respectively). Finally, *Acinetobacter* and *Enterobacter* species were resistant to ceftriaxone (66.7–68.2%) while being moderately to highly sensitive to the other antibiotics (Table 5).

## 4 Discussion

DFU is expected to become a major diabetes-associated problem in Kuwait, as in many other countries, due to the high prevalence of diabetes in the population. However, only limited data on the subject is available in the country [13, 15]. In this paper, we report estimates of DFU classification and types of DFU infection from our diabetes-dedicated institute for a period of 7 years. Our main findings were: (i) DFU was twice more likely in men than in women in our clinics, and more often present in younger men than in women; (ii) despite being younger, more males present with deeper and complex DFU than females; (iii) only about half of the

DFU were clinically infected (49.3%) but 92% of DFU showed bacterial growth in the microbiological lab analysis; (iv) comparable global findings of Gram-positive and Gram-negative strains, but significantly more Gram-positive strains are present in DFU without ischemia while more Gram-negative strains are present in DFU with ischemia; and (v) while *Staphylococcus aureus* was common in infected DFU without ischemia, *Pseudomonas aeruginosa* was more predominant in DFU with infection and ischemia, regardless of ulcer depth.

Our study population contained twice as many men as women with DFI, yet, our clinics have equal access to both sexes. This may be partially explained by a previous report which found that male sex and poor glycemic control are independent risk factors for DFI [27]. Similar trends have been observed in other countries, and authors have suggested that men are more likely to work outdoor which ultimately increases the risk to foot trauma and injury [28]. In addition, DFU were more often present in younger men than in women and despite being younger, more males presented with deeper and more complex DFU than females. Furthermore, we have observed an increased percentage of DFI in men at earlier ages than in women even without reaching statistical significance. This may be explained by the tendency of women to take more responsibility for medical care and personal hygiene, even though other factors cannot be ruled out. This hypothesis is further supported by our observations that (i) a higher percentage of women was present in the controlled HbA1c group (19.41% versus 11.95% in men) and (ii) more women in the controlled HbA1c group displayed non-infected wounds; grades A and C (43.90% and 60.6% for men and women, respectively) (Table 2 and Fig 1A).

Globally, the number of isolates found in DFI varies widely. Polymicrobial infection rates have been found to be more prevalent in studies from a number of different countries including 83.7% in Portugal [29], 60% in South India [19], 55.7% in Mumbai, India [30], and 40% in Italy [31], whereas others have reported higher rates of monomicrobial infections [32, 33]. Nevertheless, this variation in prevalence could be also related to sampling and processing techniques in each study. Overall, we found a higher percentage of DFI with monomicrobial isolates (57.3%) compared to polymicrobial isolates (34.8%) (1.30 pathogens per patient on average) and 8.2% with no growth. After excluding the no-growth samples, the average pathogens per patient increased to 1.43. Conversely, in Kuwait, two older hospital-based studies have reported high rates of polymicrobial infection (75% and 64%) in studies including 440 [15] and 86 [13] patients with DFI. This shift in number of isolates may be attributed to changes in DFU management, through regular follow-up and better patient education and knowledge throughout the country in the recent year. In Addition, DFI microbial etiology can change due to variations in healthcare systems and standard protocols as well as ethnicity and health status of the patients studied. In our study, and in agreement with a previous report from a teaching hospital in Kuwait [15], both Gram-positive and Gram-negative pathogens were similarly frequent in clinically infected DFU and no difference was observed in this pattern between sexes.

In the initial microbiological analysis of our DFI patients, we included both those with and without clinical infection as per the practices we had at the time in our institute. About half did not present with clinical signs of infection, these findings are similar to a previous large multicenter study of 1229 new DFU which reported an infection rate of 58% [34]. When performing microbiology culture analysis in our non-clinically infected patients, around 45% showed bacterial growth including at least one of the following bacteria, *Staphylococcus aureus*, MRSA, and *Pseudomonas aeruginosa*. Nevertheless, this bacterial presence was considered as colonization rather than infection. It is worth noting that following our current study and other unpublished data, the procedure has changed in our institute and now cultures are only collected from DFU with clinical signs of infection. This change in procedure is expected to reduce the costs involved in the management of DFU.

Interestingly, in our current analysis we found a significant trend between DFU depth and bacteria groups, where superficial infected DFU not involving tendon, capsule or bone and DFU penetrating to tendon or capsule were more likely to be infected by Gram-positive bacteria than those of the same depth and further complicated by the presence of ischemia. Accordingly, *Staphylococcus aureus* and MRSA were predominantly found in grades B while *Pseudomonas aeruginosa* was predominant in grade D wounds with underlying peripheral arterial disease. No association was, however, found between the number of isolates present and the DFU grade. We further assessed the potential association between DFU depth and the type of pathogens. However, even though DFU penetrating to bone and joint had a higher proportion of MRSA in comparison to wounds with less depth, the limited number of samples in some sub-groups did not allow us to draw a clear conclusion on this key question. In this context, different bacterial profiles have been reported with different wound grades, in which aggravated wounds and infections were associated with increased Gram-negative species, especially *Pseudomonas aeruginosa* [35]. However, debate remains open regarding the impact of polymicrobial infections on wound severity, as they have been reported to be more common in moderate wounds [35], while other reports suggest that polymicrobial infection increased in line with wound grading [36]. In a follow-up study over 10 weeks, Gardiner *et al.* examined DFI microbiome and found that it differed significantly and had a reduced diversity compared to that of control non-DFU wounds, and that this difference was not driven by the most abundant members of the wound pathogen community [37]. Those results and others, including our own, highlight the need for further detailed and goal-oriented designed studies to establish the potential association between strains and DFU grading before translating these debatable findings to clinical practice.

The observation that *Staphylococcus aureus* is the most common pathogen (19.9% excluding 5.0% MRSA) in our DFU patients followed by coagulase-negative *Staphylococcus* (15.7%) is concordant with most previous studies, including those from Kuwait, which also reported a high frequency of *Staphylococcus aureus* in DFI. This latter species causes the majority of human infections and has proven to be among the most persistent pathogens in healthcare facilities worldwide [38]. The MRSA rate in study is also comparable to the range reported in other countries, although prevalence rates vary widely [39, 40]. The reduction in polymicrobial infections in our study, in comparison to previous reports from Kuwait [13, 15], may reflect an improvement of healthcare system and in patients' hygiene. However, we cannot rule out other potential reasons affecting this rate, such as the younger age of our studied population. It is noteworthy that in our study 2.5% of *Staphylococcus aureus* isolates have shown resistance to Linezolid. Despite most of *Staphylococcus aureus* isolates including MRSA, are susceptible to linezolid, the emergence of linezolid-resistant MRSA has been, however, previously reported [41, 42]. While *E. coli* and *Klebsiella* species were 100% sensitive to Carbapenems, *Pseudomonas aeruginosa* was relatively less sensitive to this class of antibiotics (88.4% and 85% sensitivity, respectively to Meropenem and Imipenem). These results are comparable to a previous report from Kuwait educational hospital where *Pseudomonas aeruginosa was* 10% resistance to Imipenem [15].

On the other hand, the Gram-negative *Pseudomonas aeruginosa* was identified in 12.8% of cases as compared to previous reports from Kuwait (17.4% and 17.5%) [13, 15] which appeared higher than those in other geographical regions [43]. Interestingly, *Pseudomonas aeruginosa*, a multi-drug-resistant organism, has been reported to be highly prevalent in DFI patients from India in many studies [19, 30, 44]. These observations on the prevalence of *Pseudomonas aeruginosa* suggest regional variability probably due to microbial environments and healthcare practices such as sample collection and processing. Finally, and after dividing the data (DFU classification or infection) according to individual HbA1c levels (controlled or uncontrolled) we could not identify specific trends or associations between HbA1c levels and

types of isolates or wound grades, despite the higher rate of men with controlled HbA1c and infected wounds.

Altogether, and in the context of the alarming worldwide increase both in the prevalence of DFI and antimicrobial resistance, our results are in favor of accurate and prompt identification of DFI to ensure the responsible isolates are targeted at earliest with the appropriate antibiotics, thus limiting the unnecessary use of broad-spectrum empirical antibiotics.

Despite the relatively large number of patients and the long period of time covered in our study, some limitations need to be highlighted. One unavoidable pitfall is the fact that our samples are a mixture of swabs and tissue samples that may not give consistent results, in particular with regards to aerobic and anaerobic pathogens as we did not include any condition or analysis of the latter group. In this context, using whole genome sequencing (WGS) approach may provide a more accurate representation of microorganism profiles including non-culturable organisms and a better estimate of the microorganism "load" in DFU. Furthermore, in such studies, only DFU with clinical infection are usually investigated at the microbiological level whereas here we initially included both DFU with and without clinical infection. A standardized procedure will help improve the consistency of microbial quantitation. Indeed, the implementation of guidelines of the International Working Group on the Diabetic Foot for obtaining specimens for culture from patients with DFI has been shown to be have cost-saving potential and provided interesting quality indicators in the global management of DFI [45]. In that study, within a period of 4 years, there was a significant decrease in the number of bacterial species per sample (from 4.1 to 1.6) and in the number of MRSA cases (from 52.2% to 18.9%). Another limitation is the relatively small number of patients in some wound classification groups, which did not allow us to analyze the data to find associations between various pathogens and the wound grading. Finally, our study was retrospective and used only UoT DFU classification system, therefore, a follow-up cohort study with these patients is needed to shed light on potential prediction of clinical outcomes, including wound healing in relation with the type of pathogens, DFU classification, clinical management, and the control of diabetes.

In conclusion, we provided an updated picture of the DFI pattern and showed an improved DFI pattern in comparison to the results in the previous reports from Kuwait and the region. Clinicians should keep up to date on common pathogen patterns in their local area, due to vast regional variability especially when prescribing empirical antibiotics. A combined DFI grading with microbial profile and sensitivity tests will help guide clinicians on the appropriate treatment and wound management.

## Supporting information

**S1 Table. Minimal data set for study population characteristics.**
(XLSX)

## Acknowledgments

We would like to thank the staff at the Clinical Laboratory, Dasman Diabetes Institute, for their assistance throughout this study. The authors would like to thank Enago (www.enago. com) for the English language review.

## Author Contributions

**Conceptualization:** Asma Alhubail, May Sewify, Ali Tiss.

**Data curation:** May Sewify, Grace Messenger, Richard Masoetsa, Shinu Nair, Ali Tiss.

**Formal analysis:** Asma Alhubail, May Sewify, Grace Messenger, Ali Tiss.

**Investigation:** Asma Alhubail, May Sewify, Grace Messenger, Richard Masoetsa, Imtiaz Hussain, Shinu Nair, Ali Tiss.

**Methodology:** Asma Alhubail, May Sewify, Grace Messenger, Richard Masoetsa, Imtiaz Hussain, Ali Tiss.

**Project administration:** Ali Tiss.

**Resources:** Asma Alhubail, May Sewify, Grace Messenger, Imtiaz Hussain.

**Software:** May Sewify.

**Supervision:** Asma Alhubail, Ali Tiss.

**Validation:** Asma Alhubail, May Sewify, Grace Messenger, Richard Masoetsa, Imtiaz Hussain, Ali Tiss.

**Writing – original draft:** May Sewify, Ali Tiss.

**Writing – review & editing:** Asma Alhubail, May Sewify, Grace Messenger, Richard Masoetsa, Imtiaz Hussain, Shinu Nair, Ali Tiss.

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
