## [Decision Letter · Decision Letter 0]

24 Aug 2020

PONE-D-20-08931

Microbiological profile of diabetic foot ulcers in Kuwait

PLOS ONE

Dear Dr. Tiss,

Thank you for submitting your manuscript to PLOS ONE. After careful consideration, we feel that it has merit but does not fully meet PLOS ONE’s publication criteria as it currently stands. Therefore, we invite you to submit a revised version of the manuscript that addresses the points raised during the review process.

Several concerns are shared by the reviewers and myself. The design and results of the study need to be further detailed and need to be contextualized with previous work. Also, some conclusions drawn are not fully supported and several observations should me more thoroughly discussed in view of the current global antibiotic resistance situation. Furthermore, the statistical analysis and the interpretation should of the data should be thoroughly revised according to the suggestions of the reviewers. The authors are invited to include further information and the recommended controls and to provide a revised manuscript and a point-by-point response to each comment raised by the reviewers.

We look forward to receiving your revised manuscript.

Kind regards,

Rita G. Sobral, PhD

Academic Editor

PLOS ONE

Journal Requirements:

Reviewers' comments:

Reviewer's Responses to Questions

**Comments to the Author**

1. Is the manuscript technically sound, and do the data support the conclusions?

Reviewer #1: Partly

Reviewer #2: No

Reviewer #3: Yes

2. Has the statistical analysis been performed appropriately and rigorously? 

Reviewer #1: No

Reviewer #2: No

Reviewer #3: Yes

3. Have the authors made all data underlying the findings in their manuscript fully available?

Reviewer #1: No

Reviewer #2: Yes

Reviewer #3: Yes

4. Is the manuscript presented in an intelligible fashion and written in standard English?

Reviewer #1: Yes

Reviewer #2: Yes

Reviewer #3: Yes

5. Review Comments to the Author

Reviewer #1: While the topic of the manuscript is indeed interesting and relevant, I have a set of concerns/queries, as outlined below:

- The first cited reference is a previous study published by part of the authors also involved in the current manuscript: “Messenger G, Masoetsa R, Hussain I, Devarajan S, Jahromi M: Diabetic foot ulcer outcomes from a podiatry led tertiary service in Kuwait. Diabet Foot Ankle 2018, 9:1471927.” Much of the same methodology was used, but the first paper looked at different outcomes, in the time span 2014-2016. The following microbiology data is reported in this previous article: “Microbiology samples were collected from 150 DFUs, of which nearly 40% were positive for bacterial growth. The most common microorganisms were Gram-negatives (16%), and 11 (3%) new cases of Methicillin-resistant Staphylococcus aureus (MRSA) were identified.” I find it hard to understand why a new paper would be needed, and why publish these two analyses as two separate manuscripts.

- Line 24: The phrasing should be revised, since it is an overstatement to say that the study’s aim was to characterize DFU “within Kuwait”. The same can be said for the conclusions on line 40 “patterns in Kuwait”, and for the manuscript’s title. This data reflects, at best, the population attending the DDI center in Kuwait in the study time span.

- Furthermore, this study population attending the DDI center should be clearly defined on line 93, where a brief description of the hospital is first given.

- Line 26: A brief statement regarding the criteria used for selecting patients for the study should be included here. For example: all consecutive patients attending the Institute within a specific time span.

- Lines 38 and 39: It would be easier to follow the numbers if the same order is used for reporting. In these phrases female cases are reported first and then male cases first.

- Line 41: “the observed regional variability of DFI pathogens” – this study has not looked at regional variability specifically, only the discussion section provides a comparison with different regions/countries. However, this comparison cannot truly reflect a regional variability, since differences in methodology can be reasonably expected.

- Line 42: Since this was not a study of wound healing time/quality, or of clinical outcomes, the following phrase is not a direct conclusion of the study: “to enhance wound healing and improve clinical outcomes.”

- Line 49: “The lifetime risk of developing a DFU is estimated to be 12 to 25% [1, 2].” This phrase requires clarification. Are the findings from literature reported here valid only for patients with diabetes? Or for the general population? Furthermore, it is not clear from which literature reference this information was derived, since the reference quoted here (#1) cites the following different information, for the same reported percentages, and further citing another literature sources: “Diabetic foot ulcers (DFUs) continue to be a leading cause of non-traumatic lower limb amputation [1], with an estimated lifetime risk of 12–25% [2–4].” Citations should generally point towards the initial source and it should be checked that the intended meaning is portrayed.

- Line 53: “Despite the fact that the prevalence of diabetes in Kuwait is ranked among the highest worldwide” – how high?

- Line 64: Please recheck italicizing and capitalization. All bacterial genus and species names should be written in italic font, whereas “spp.” or “species” should not. Genus names should be capitalized, while species names should not.

- Lines 72-73: Depth of the wound and sampling technique can also be involved.

- Line 74: The term “colonizers” should be used only when talking about colonization, i.e., not infection.

- Line 100 and ethics statement: The Declaration of Helsinki generally refers to the rights of subjects in interventional clinical trials. Also, a statement should be included regarding the waiver of informed consent by an ethics committee.

- Line 128: CAP should be defined at the first use in the text.

- Line 149: Check spelling and italicization: Acinitobacter, etc.

- Line 158: Check producer for this SPSS version. It should be: IBM Corp., Armonk, NY, USA)’.

- Line 159: Student’s t test is only appropriate for variables with parametric distribution. Was variable distribution checked for all continuous variables?

- Lines 205, 232, 308 and other occurrences: By Staphylococcus aureus, are the authors referring to MSSA?

- Lines 226-229: Check the use of italic font.

- Line 243: Correlation analysis is only applicable for continuous variables. An “association” should be discussed instead.

- Lines 245 and all antimicrobial names: Check capitalization.

- Line 248: The 97.5% susceptibility to linezolid should be discussed as this is an important, worrisome finding.

- Line 252: The decreased susceptibility to carbapenems should be discussed.

- Figure 1: “Controlled male/female” and “Uncontrolled male/female” sounds a bit awkward and should be revised. Furthermore, a figure legend should clarify that this term refers to glycemic control and the values for control should be stated here, i.e., 6.5.

- Full statistical comparisons should be included in the relevant tables and only discussed in the text.

- Table 1 is hard to follow.

- Table 2, Table S1 and analysis in the manuscript text: Please revise the definition of “controlled” and “uncontrolled”. At the current moment a patient with 6.5 does not fit into any of these categories. Please revise and recalculate.

- Table S1: This table is not very relevant. Furthermore, in the A3 rows, how can a “controlled” patient have an HBa1c level > 10?

- Table 3 is not needed, as much of this information is already presented in the text.

- Table 4: Please revise the use of italic font and spelling of some of the bacterial names.

- Table 5 is a bit hard to follow because of the different column names in the first and second half of the table.

- Table 5: Data on cefoxitin screening for S. aureus should also be included.

- The references used for comparison in the Discussion section should overall be read more carefully, to avoid misinterpretation of the data when citing their results.

- Line 271: These are not “the first estimates”, as the authors have already published data on this topic, as mentioned above.

- Lines 286-289: When discussing mono- vs poly-microbial infection, it would be important to look at the sampling and processing techniques in each study, otherwise the overall numbers cannot be compared.

- Line 340: It is not clear what this phrase refers to: “the most persistent pathogens in healthcare facilities worldwide”.

- Lines 341-344: The authors state that: “The presence of MRSA in our study was only 5.0% of the total isolates detected. This rate is much lower than European statistics. Recently, The European Centre for Disease Prevention and Control (ECDC) reported that the prevalence of MRSA decreased across Europe from 19.6% in 2014 to 16.9% in 2017 [43].” However, these numbers cannot be compared, since the cited report reflects the percentage of MRSA isolates from the total number of S. aureus isolates, while the percentage reported in the current manuscript does not.

- Line 345: Reduction compared to what?

- Line 346: Improvement compared to what/when?

- Line 354: What differences in healthcare practice?

- The reference list is quite long for an original article, and many of the references included here are either outdated, or not directly relevant to the current study.

- Data availability statement: “All generated data and resources are reported in this manuscript and there is no other data to be provided.” The source dataset should also be made available.

- English language is overall ok, but it should be revised to address occasional mistakes or errors.

Reviewer #2: I would like to thank the authors for their paper. The topic of diabetic foot ulcers and the microbiological profiles will be of great interest to readers. While the paper is well-written, I believe there is a major methodological error in this study, which inhibits the ability to draw valid conclusions. Please refer below to my full list of thoughts, comments, suggestions, and questions:

• Lines 148-156: Materials and Methods: Antimicrobial Susceptibility Testing- Please include the antibiotic list and concentrations used. Adding the concentrations of discs used will ensure reproducibility.

• Line 208: Results: Microbial Isolate Etiologies in DFUs with clinical infection- The authors indicate that to characterize the microbiological profile they only analyzed “clinically infected wounds”. The “clinically uninfected” could essentially have served as the control group. When characterizing microbiological profiles, it is very essential to have a control group that would help understand the “baseline microbiological composition”, and thus would help make valid conclusions about what truly is the microbiological composition of the diseased state (in this case clinically infected samples). Not characterizing the microbiological composition of a control is a major methodological error of this study.

• Line 244: Results: Antimicrobial susceptibility pattern- Please switch the outcome to % resistant instead of %sensitive/susceptible. Since the authors report % resistance for Acinetobacter and Enterobacter species, I believe it would be beneficial if the authors continue to report the results of the antimicrobial resistance patterns as % resistant.

• Discussion: Please include the significance of the antimicrobial resistance testing results in the discussion and what this would mean for the clinical management of DFUs and DFIs.

• Discussion: I would also recommend the authors to include another limitation of the study in their discussion. Since the authors have employed a culture-based approach to characterize microorganisms, some microorganisms might be unculturable. Using whole genome sequencing (WGS) to characterize microbiological profiles may provide a more accurate representation of microorganism profiles. Not only will WGS characterize non-culturable organisms, but the number of reads can be used to estimate the “load” of microorganisms in DFUs.

• Line 557 Table S1: I would recommend the authors label each stage and grade and depth for clarity. For instance, label A0 with “Pre- or post-ulcerative lesion completely epithelialized”, A1 with “Superficial wound not involving tendon, capsule or bone” so on and so forth.

Reviewer #3: Microbiological profile of diabetic foot ulcers in Kuwait

This paper describes a retrospective study of the microbiology profile diabetic foot ulcers in Kuwait. It adds little if any novel findings to the field; however, such is the nature of studies of this type and that should not preclude publication.

Overall, the paper is well written and well-presented. However, there are several major and minor revisions which we need to be addressed before publication.

Major revisions

1. It would be useful if the authors could present a breakdown of the number of patients whose ulcers (x/260) or infections (x/253) were investigated using either swab or tissue sample collection.

The authors are correct in their discussion when they highlight that this is a limitation of this work. It may also reflect the high proportion of coagulase-negative staphylococcus identified in table 4, and the potential for contaminating skin flora to be detected by superficial swabs should be highlighted in the discussion.

2. This paper should include a clear breakdown of the number of bacterial isolates detected from the 253 infected patients, regardless of their Texas grade, and the frequency with which each bacterial species was identified. This should come in table format.

Minor revisions

1. There are multiple instances in this paper where the abbreviation for diabetic foot infection or diabetic foot ulcer is used incorrectly. The authors need to carefully check that every time they have used a three-letter acronym (DFU vs. DFI) they have used the correct one. Examples of lines where attention is required include:

31-‘DFU infections’

80-‘microbes isolated from DFIs from a different DFUs using the University of Texas…’ This needs clarifying, I think you mean provide power microbes isolated from DFI’s from different ulcer grades.

82-should be DFI pathogens

83-should be the microbial profile of DFIs

112

211-an infected DFU is a DFI

These examples are not exhaustive, there may be others.

2. Throughout the manuscript you have used sex and gender interchangeably. Please use sex throughout.

3. Please define the abbreviation CAP-line 128

4. There is a formatting issue on lines 192 and 201-please correct. Also on these lines, please insert ‘df’ if you refer to degrees of freedom in the parentheses. At it’s first use degrees of freedom should be defined as ‘df’.

5. Line 193-please write ‘years’ in full

6. Line 202. If you have used an ‘F test’ please also include this in your methods.

7. Line 209-please remind the reader of the number of clinic infected wounds to which you are referring – e.g. (n = x)

8. Throughout the manuscript you have been inconsistent in your spacing of figures. For example, n=x vs. n = x. Please choose one format and stick with it throughout.

9. Line 248 please insert ‘was’ between ‘but less’

10. Line 272, should read ‘contained twice as many men as women’

11. Line 290, the claim that most diabetic foot infections are ‘dominated’ by Gram-negative strains is based on five references here. All these references come from papers published in India or China. You cannot make a statement about ‘most reports’ easier handfuls of papers from very limited geographical sources. As it happens, certainly in Western Europe and the United States, it is most often gram-positive cocci which predominate in these infections. I suggest removing this sentence.

12. Line 305, extra space has been inserted

13. line 339, you mean species not strain.

14. Table 1, please sort out formatting, move the heading ‘controlled’ up

15. Table 2, please clarify for the reader that you mean Texas wound grading. This needs to be clarified in both the legend and the table.

16. Figure 2, do you mean species instead of strain?

6. PLOS authors have the option to publish the peer review history of their article (what does this mean?). If published, this will include your full peer review and any attached files.

Reviewer #1: No

Reviewer #2: **Yes: **Sanjana Mukherjee

Reviewer #3: No

---

## [Author Response · Author response to Decision Letter 0]

11 Nov 2020

Please see attached detailed response to Reviewers

---

## [Decision Letter · Decision Letter 1]

8 Dec 2020

Microbiological profile of diabetic foot ulcers in Kuwait

PONE-D-20-08931R1

Dear Dr. Tiss,

We’re pleased to inform you that your manuscript has been judged scientifically suitable for publication and will be formally accepted for publication once it meets all outstanding technical requirements.

Kind regards,

Rita G. Sobral, PhD

Academic Editor

PLOS ONE

Additional Editor Comments (optional):

Reviewers' comments:

Reviewer's Responses to Questions

**Comments to the Author**

1. If the authors have adequately addressed your comments raised in a previous round of review and you feel that this manuscript is now acceptable for publication, you may indicate that here to bypass the “Comments to the Author” section, enter your conflict of interest statement in the “Confidential to Editor” section, and submit your "Accept" recommendation.

Reviewer #1: All comments have been addressed

Reviewer #4: All comments have been addressed

2. Is the manuscript technically sound, and do the data support the conclusions?

Reviewer #1: Yes

Reviewer #4: Yes

3. Has the statistical analysis been performed appropriately and rigorously? 

Reviewer #1: Yes

Reviewer #4: Yes

4. Have the authors made all data underlying the findings in their manuscript fully available?

Reviewer #1: Yes

Reviewer #4: Yes

5. Is the manuscript presented in an intelligible fashion and written in standard English?

Reviewer #1: Yes

Reviewer #4: Yes

6. Review Comments to the Author

Reviewer #1: (No Response)

Reviewer #4: (No Response)

7. PLOS authors have the option to publish the peer review history of their article (what does this mean?). If published, this will include your full peer review and any attached files.

Reviewer #1: No

Reviewer #4: No

---

## [Editor Report · Acceptance letter]

11 Dec 2020

PONE-D-20-08931R1 

Microbiological profile of diabetic foot ulcers in Kuwait 

Dear Dr. Tiss:

I'm pleased to inform you that your manuscript has been deemed suitable for publication in PLOS ONE. Congratulations! Your manuscript is now with our production department. 

Kind regards, 

on behalf of

Dr. Rita G. Sobral 

Academic Editor

PLOS ONE